

# Climate information preserved in seasonal water isotope at NEEM: relationships with temperature, circulation and sea ice

Minjie Zheng[1*], Jesper Sjolte[1], Florian Adolphi[1,2], Bo Møllesøe Vinther[3], Hans Christian Steen-Larsen[4], Trevor James Popp[3], and Raimund Muscheler[1]

[1]Department of Geology, Quaternary Science, Lund University, Lund, Sweden

[2]Climate and Environmental Physics, Physics Institute, and Oeschger Centre for Climate Change Research, University of Bern

[3]Centre for Ice and Climate, Niels Bohr Institute, University of Copenhagen, Copenhagen, Denmark

[4]Geophysical Institute and Bjerknes Centre for Climate Research, University of Bergen, Norway

*Correspondence to: Minjie Zheng (minjie.zheng@geol.lu.se)*

**Abstract**

Analyzing seasonally resolved $\delta^{18}O$ ice core data can aid the interpretation of the climate information in ice cores, providing also insights into factors governing the $\delta^{18}O$ signal that cannot be deciphered by investigating the annual $\delta^{18}O$ data only. However, the seasonal isotope signal has not yet to be investigated in northern Greenland, e.g. at the NEEM (North Greenland Eemian Ice Drilling) ice core drill site. Here we analyze seasonally resolved $\delta^{18}O$ data from four shallow NEEM ice cores covering the last 150 years. Based on correlation analysis with observed temperature, we attribute about 70% and 30 % of annual accumulation to summer and winter respectively. The NEEM summer $\delta^{18}O$ signal correlates strongly with summer western Greenland coastal temperature and with the first principal component (PC1) of summer $\delta^{18}O$ from multiple seasonally resolved ice cores from central/southern Greenland. However, there are no significant correlations between NEEM winter $\delta^{18}O$ data and western Greenland coastal winter temperature, or southern/central Greenland winter $\delta^{18}O$ PC1. The stronger correlation with temperature during summer and the dominance of summer precipitation skew the annual $\delta^{18}O$ signal in NEEM. The strong footprint of temperature in NEEM summer $\delta^{18}O$ record also suggests that the summer $\delta^{18}O$ record, rather than the winter $\delta^{18}O$ record, is a better temperature proxy at the NEEM site. Despite dominant signal of North Atlantic Oscillation (NAO) and Atlantic Multidecadal Oscillation (AMO) in the central-southern ice cores data, both NAO and AMO exert weak influences on NEEM seasonal $\delta^{18}O$ variations. The NEEM seasonal $\delta^{18}O$ is found to be highly correlated with Baffin Bay sea ice concentration (SIC) in satellite observation period (1979-2004), suggesting a connection of the sea ice extent with $\delta^{18}O$ at NEEM. NEEM winter $\delta^{18}O$ significantly correlates with SIC even for the period prior to satellite observation (1901-1978). The NEEM winter $\delta^{18}O$ may reflect sea ice variations of Baffin Bay rather than temperature itself. This study shows that seasonally resolved $\delta^{18}O$ records, especially for sites with seasonal precipitation bias such as NEEM, provide a better understanding of how changing air temperature and circulation patterns are associated with the variability of the $\delta^{18}O$ records.





## 1. Introduction

Stable water isotopes in Greenland ice cores, e.g. $\delta^{18}O$, provide key information on temperature (Küttel et al., 2012), moisture source (Masson-Delmotte et al., 2005b), sea ice extent (Noone and Simmonds, 2004) or atmospheric circulation (Vinther et al., 2003). The available data have revealed the complexity of the integrated information preserved in the stable water isotope composition of Greenland ice cores (Masson-Delmotte et al., 2005a), thereby illustrating the need for improving our understanding of its climatic controls. Recent studies indicate that having not only the annual, but seasonally resolved ice core $\delta^{18}O$ data, represents a significant improvement for the interpretation of the $\delta^{18}O$ signal (Vinther et al., 2003;Vinther et al., 2010). For example, Ortega et al. (2014) indicated that the seasonal $\delta^{18}O$ records allow to reconstruct the variability of weather regimes in the North Atlantic region.

Vinther et al. (2010) extracted the seasonal $\delta^{18}O$ from 13 ice cores in central and southern Greenland. However, seasonally resolved data are still lacking from northern Greenland, for example, the NEEM (North Greenland Eemian Ice Drilling, 77.45° N, 51.06° W, 2450 m a.s.l., Fig. 1) ice core. The NEEM project originally aims to retrieve an ice core record spanning the last interglacial period (Neem community members, 2013). To assist interpreting the stable isotope record along the deep ice core, several shallow firn/ice cores were also drilled around the camp as part of the exploration program. Through investigating these short cores, the results suggest that the NEEM annually resolved $\delta^{18}O$ records correlate unexpectedly weakly to the annual and winter North Atlantic Oscillation (NAO) signal (Steen-Larsen et al., 2011;Masson-Delmotte et al., 2015). This contrasts with $\delta^{18}O$ records from central and southern part of Greenland that strongly correlate with the winter NAO signal (Vinther et al., 2003;Vinther et al., 2010). Regional to global atmospheric models show that precipitation at NEEM is dominated by summer precipitation, which may contribute to the lack of the winter NAO fingerprint in annual NEEM $\delta^{18}O$ records (Steen-Larsen et al., 2011). This seasonal precipitation bias may skew the annual $\delta^{18}O$ signal towards summer precipitation and cause a weak correlation to the NAO which exerts its strongest influence on Greenland weather in winter. Indeed, there is no explanation yet for the strong correlation between the first principal component (PC1) of 16 annually resolved Greenland $\delta^{18}O$ records and NEEM annual $\delta^{18}O$ records despite the missing NAO fingerprint in NEEM data (Masson-Delmotte et al., 2015). Furthermore, Steen-Larsen et al. (2011) found that the annual sea ice extent anomaly in the Baffin Bay explains up to 34% of variations of the annual NEEM $\delta^{18}O$ record. Hence, studying seasonally resolved NEEM $\delta^{18}O$ might help us to explore the possible seasonal relationship with the Baffin Bay ice concentration.

In this study, we follow the approach of Vinther et al. (2010) to extract the winter and summer $\delta^{18}O$ signal from four NEEM short cores. To reduce noise, the records are averaged for the overlap period from 1855 to 2004 C.E. We then compare the seasonal $\delta^{18}O$ NEEM record with other seasonal $\delta^{18}O$ records from central and southern Greenland and their first principle component (PC1) (Vinther et al., 2010). Meteorological parameters like temperature and sea level pressure are also compared with the NEEM seasonal $\delta^{18}O$ data to explore temporal and spatial relationships. The Baffin Bay sea ice concentration (SIC) data covering both the satellite period (1979-2004) and the period prior to satellite observation (1901-1978), are also compared with NEEM $\delta^{18}O$ data. The aim is to identify the seasonal $\delta^{18}O$ signal at NEEM and to investigate which parameters control the NEEM $\delta^{18}O$ variations for each season in terms of seasonal weather/climate variability.





### 2. Meteorological data

### 2.1 Temperature records

The length of observational records and locations of meteorological stations are crucial for a robust correlation between ice cores and meteorological observations. The Pituffik station is the only observation station in the northwestern part of Greenland (NW Greenland) and the closest one to the NEEM site (Fig.1; Cappelen, 2017). Although the temperature record only covers the period back to 1948, the Pituffik station is the best source of information on the weather and climate in NW Greenland. As the ice core data spans the last 150 years, we also test our $\delta^{18}O$ record against longer-term temperature observations from southwestern part of Greenland (SW Greenland). The SW Greenland temperature record is a merged temperature dataset based on 13 observational records along the southwestern Greenland costal area spanning the period 1784-2005 (Fig.1; Vinther et al., 2006). This data set covers the complete period of seasonally resolved ice core isotope data from NEEM facilitating an extended comparison period. The changes in NW Greenland costal temperatures are regionally consistent around western costal Greenland (Hanna et al., 2012;Wong et al., 2015). Therefore, some consistency of the SW Greenland temperature record with temperatures closer to NEEM can be expected.

### 2.2 Twenty Century Reanalysis data

The Twenty Century Reanalysis (20CR; Compo et al., 2011) data set is selected to investigate the relationship between NEEM isotope records and atmospheric circulation patterns and temperature. The 20CR data is a global atmospheric 2 by 2 degree gridded climate model dataset only assimilating surface observations of synoptic pressure, and using sea surface temperature and sea ice concentration as boundary conditions (Compo et al., 2011). This dataset provides estimates of global atmospheric variability spanning 1851 to 2012 at six-hourly resolution. However, there are very few stations delivering pressure data over the Greenland area until 1922 after which the number of observation stations increases significantly (Compo et al., 2011). This leads to a less well-constrained reanalysis data set for the Greenland for the period before 1930. To test the results for the early period, we divide the whole period into two subperiods 1855-1930 and 1931-2004 and examine correlations to ice core data within these subperiods. The aim is to investigate the influence of temperature and atmospheric circulation on NEEM seasonal $\delta^{18}O$ signals.

### 2.3 Indices of climate patterns

Previous analyses have related the variability in the Greenland ice core stable water isotopes to changes in the atmospheric North Atlantic Oscillation (NAO; Barlow et al., 1993;Vinther et al., 2003) and the oceanic Atlantic Multidecadal Oscillation (AMO; Chylek et al., 2012). In this study, these two indices are extracted from the 20CR dataset. We choose the PC-based NAO (NAOPC) indices which optimally represents the NAO pattern spatially and temporally (Hurrell and Deser, 2009). To obtain the monthly NAOPC index, we performed the empirical orthogonal function (EOF) on monthly pressure anomalies over the Atlantic sector, 20°-80°N, 90°W-40°E. The leading mode of EOF is used as the monthly NAOPC index. For the AMO index, we first average the sea surface temperature anomalies over the sector 0°-60°N, 0°-80°W then subtract the average sea surface temperature anomalies between 60°S-60°N from it (Trenberth and Shea, 2006). By calculating indices from the 20CR data, both indices can cover the period 1855-2004.





**2.4 Baffin Bay ice concentration**
Steen-Larsen et al. (2011) suggested a strong link between annual sea ice cover in Baffin Bay and NEEM
annual $\delta^{18}$O signal. To test this hypothesis, we selected the COBEsic sea ice data set to compare with the NEEM
seasonal $\delta^{18}$O data. The COBEsic record (Hirahara et al., 2014) is a combination of monthly globally complete
fields of sea ice concentration on a 1 by 1 degree grid based on satellite observation starting after 1979 and
historical data provided by Walsh and Chapman (2001). The mean Baffin Bay area sea ice concentration was
calculated by averaging the values over the area between 65-80° N and 80-50° W (Tang et al., 2004).
**3 Ice core data**
**3.1 The NEEM shallow ice core data**
The annual $\delta^{18}$O data from four shallow NEEM ice cores (NEEM07S3; NEEM08S2; NEEM08S3;
NEEM10S2) have been published by Masson-Delmotte et al. (2015). The shallow cores cover depths ranging
from the surface down to between 52.6 and 85.3 m. A back-diffusion calculation following Johnsen et al. (2000)
was applied to the $\delta^{18}$O records to restore the original variability and hence, improve the identification of
individual years. The annual dating of those records was performed by counting the seasonal cycles in $\delta^{18}$O and
verified by identifying signals of volcanic eruptions in the electrical conductivity measurements (Masson-
Delmotte et al., 2015). The four shallow cores share a common period from 1855-2004 which is focus in this
study.
**3.2 Greenland seasonal $\delta^{18}$O data**
The NEEM seasonal $\delta^{18}$O data are also compared with other seasonal records obtained from 13 sites in
central and southern Greenland over the period 1778-1970 (Fig.1; Vinther et al., 2010). Most records originate
from single ice core while some are stacked records from multiple cores (GRIP, n=6; DYE3-71/79, n=2). The
first principal component (PC1) of these ice core data is considered as representative of the seasonal $\delta^{18}$O signal
of central and southern Greenland. Vinther et al. (2010) divided the Greenland seasonal $\delta^{18}$O data into summer
and winter season corresponding to May-Oct and Nov-Apr, respectively.
**4 The definition of seasonal $\delta^{18}$O data**
To classify the seasons, we assume that the extremes in the seasonal cycle of the $\delta^{18}$O data correspond to
the intra-annual temperature extremes (Vinther et al., 2010). According to the SW Greenland and Pituffik
temperature records, summer temperature maxima and winter temperature minima usually occur in July/August
and January/February, respectively. For summer, we assign the maxima $\delta^{18}$O within the selected year to
July/August. For winter, the mid-winter is already defined as the onset of the annual layers by Masson-Delmotte
et al. (2015) based on the analysis of a combination of ice core data. Based on their time scale, we define onset
of the annual layer (mid-winter) to January/February. Here, we only investigate the winter and summer season as
it is very hard to reliably pinpoint the spring and autumn in the $\delta^{18}$O record. Another essential prerequisite for the
classification of seasons is the sufficient accumulation rate to guarantee a clear preservation of the seasonal cycle
(no less than 20 cm ice accumulation per year; Johnsen et al., 2000). At NEEM the estimated accumulation rate
is 21.6 cm yr$^{-1}$ for the period of 1725-2007 meeting this requirement (Gfeller et al., 2014).



The calculation of the summer mean $\delta^{18}O$ is centered around the $\delta^{18}O$ maxima value within the selected
year. For the winter mean $\delta^{18}O$ is centered around the onset of annual layer within the selected year. We then
take different fractions of annual accumulation symmetrically around the seasonal center. This is done for four
ice cores and these 4 seasonal $\delta^{18}O$ series data are averaged to minimize noise. Finally, we correlate the averaged
seasonal $\delta^{18}O$ data to the winter and summer temperatures defined with different choices of season length.

156        Fig. 2 shows the result of the correlation analysis between different choices of winter and summer

temperatures with different fractions of the NEEM annual $\delta^{18}O$ signal. For SW Greenland and Pituffik summer
temperature records (Fig. 2a and Fig. 2b), the highest correlations occur between May-October averaged
temperature and a fraction of around 70% annual accumulation. In contrast, there is no significant correlation
peak found when comparing NEEM winter $\delta^{18}O$ with different choices of winter temperatures in NW and SW
Greenland. However, it is interesting to note the correlation peak with the Pituffik temperature record in Fig. 2d
at 30% annual accumulation, although not significant, which complements the result for the summer
$\delta^{18}O$/temperature correlation. For the winter signal the most significant correlation is obtained when the annual
average SW Greenland temperature (Aug-Jul; Fig. 2c) is compared with annual average $\delta^{18}O$ data (100% of the
annual accumulation centered around the mid-winter). This significant correlation is likely due to the fact that
the annually resolved $\delta^{18}O$ includes the summer signal which indicates high correlation with annual average
temperature that includes a strong imprint of the summer temperature. Furthermore, the correlation between
NEEM winter $\delta^{18}O$ data and SW Greenland temperature shows no correlation peak which is quite different from
the one with the Pituffik record (Fig. 2d). The different relationships (Fig. 2c & Fig. 2d) suggest that the
correlation between temperature and NEEM winter $\delta^{18}O$ may vary for different periods. However, it should be
noted that the Pituffik and SW Greenland temperature records represent different parts of Greenland climate over
different time spans. We further examine the correlation between $\delta^{18}O$ and SW Greenland temperature for 1949-
2004 (Fig. S1, supplementary information). As expected, the correlation with SW Greenland over the period
1949-2004 displays similar dependencies as the one shown in Fig. 2b & 2d for the Pituffik station, supporting
the conclusion of a changing relationship between winter $\delta^{18}O$ and Western Greenland temperatures over time.
This weak and varied correlations of winter $\delta^{18}O$ and temperature can likely be attributed to the intermittent and
low winter precipitation at NEEM (Steen-Larsen et al., 2011). The correlation for SW Greenland during 1949-
2004 shows the most significant correlation at higher annual accumulation for summer (80% for Apr-Nov) and
lower for winter (peak at 20%). This result is consistent with the one indicated by Pituffik records.

180        Based on these results we conclude that, on average, about 70% of annual accumulation occurs between

May-Oct, while the remaining 30% of annual accumulation occurs during Nov-Apr. We note that irrespectively
of the actual process recording the $\delta^{18}O$ in the snow being either precipitation weighted $\delta^{18}O$, a signal only
driven by atmospheric water vapor isotopes as suggested by Steen-Larsen et al. (2014), or a combination our
conclusion would still hold. An example of the chosen definition of seasons is shown in Fig. S2 (supplementary
information). This conclusion is based on the strong and consistent correlation with two summer temperature
data sets and the correlation peak for winter shown in Fig 2d. This conclusion is further supported by the
comparison with the measured precipitation data in Pituffik station over the 1949-2000. Although the
precipitation data are incomplete (almost no available data for 1976-1993), the average ratio of summer (May-
Oct averaged) and winter (Nov-Apr averaged) precipitation over 1946-2000 is around 2 which is similar with





accumulation ratio in this study (summer/winter=2.3). This season definition also accords with seasonal
classification in central and southern Greenland (Vinther et al., 2010).
Generally, the temperature imprint on NEEM $\delta^{18}O$ is higher during summer than winter. The NEEM
summer $\delta^{18}O$, rather than NEEM winter $\delta^{18}O$, is a better temperature proxy for the NEEM site and likely for
northwestern Greenland. This result is in contrast to the finding that winter $\delta^{18}O$ records in central/southern
Greenland have been shown to be the better temperature proxy for past Greenland temperature conditions
(Vinther et al., 2010). Therefore, one should be cautious when combing the NEEM seasonal $\delta^{18}O$ with other ice
cores data for use in temperature reconstructions. Another interesting feature is the dominant summer
precipitation at the NEEM site (contributing to 70% of annual accumulation) compared to the ice cores in the
central/southern Greenland (50% of annual accumulation for both season). Even though the investigated period
only covers the last 150 yrs, knowing this seasonal variability can aid the climate interpretation of the long-term
$\delta^{18}O$ variability. For example, climate model simulations suggest that seasonality changes over time with a
decrease in winter precipitation during the glacial period, which would strongly affect sites with considerable
winter accumulation, while being potentially less important for the sites, such as NEEM, with little winter
accumulation (Werner et al., 2000).

## 5 The seasonal $\delta^{18}O$ data

### 5.1 NEEM records and signal to noise ratio

For low accumulation sites like NEEM, it is important to examine the signal to noise ratio (SNR; see
Vinther et al. (2006) for a derivation of the SNR) in the $\delta^{18}O$ data. The SNR for the $\delta^{18}O$ data in NEEM cores is
0.64 for the winter and 1.28 for the summer. The winter $\delta^{18}O$ is more strongly influenced by noise than the
summer signal possibly due to windier conditions and less snow accumulation. These two SNRs are in line with
a previous study by Masson-Delmotte et al. (2015) that found a SNR of 1.3 for the annual NEEM $\delta^{18}O$. Note that
the seasonal SNRs observed here are higher than the level obtained for six ice cores from the GRIP project (0.57
for winter and 0.89 for summer; Vinther et al., 2010). Therefore, we conclude that the set of these four ice cores
is sufficient to extract a robust seasonal $\delta^{18}O$ at NEEM.

### 5.2 Comparison with other Greenland ice core records

Fig. 3 presents the correlation between seasonal stacked NEEM $\delta^{18}O$ and other seasonal ice cores in
Greenland, including the Greenland $\delta^{18}O$ PC1. All data are detrended before correlation. The NEEM summer
$\delta^{18}O$ data are significantly correlated with the summer Greenland ice core isotope data from locations in southern
Greenland and to the west of the central ice divide (Fig. 3a; with correlation from 0.3 to 0.46). However, summer
$\delta^{18}O$ from cores located to the east of the central ice divide (Renland, Site E, G and A) do not correlate
significantly with the NEEM summer $\delta^{18}O$ data. This is in accord with the fact that moisture pathways are
different for snow accumulation to east and west of the central ice divide (Vinther et al., 2010). Therefore,
having ice core records from both east and west side of the ice divide facilitates identification of regional-scale
atmospheric variability. The correlation between NEEM summer $\delta^{18}O$ and the Greenland summer PC1 record is
significant both in inter-annual (r=0.54) and 11-year smoothed data (r=0.67, Fig. 4a and c). The correlations of
11-yr averaged data are tested using the 'Random-phase' method introduced by Ebisuzaki (1997). The
correlations are consistent with the correlation between annual NEEM $\delta^{18}O$ and Greenland $\delta^{18}O$ PC1 found by



228 Masson-Delmotte et al. (2015). NEEM winter $\delta^{18}O$ shows no significant correlation with most winter Greenland

229 $\delta^{18}O$ records, and weak negative correlation with three southern ice core records (DYE3-71/79, 18C, 20D; Fig.

230 3b). No correlations are observed for the comparison with Greenland winter $\delta^{18}O$ PC1 at inter-annual and

231 decadal scale (Fig. 4b and d). The results indicate a rather different winter climatic fingerprint archived in

232 northwestern Greenland suggesting one needs to be careful when interpreting the NEEM winter $\delta^{18}O$ records.

233 Such poor correlations between NEEM winter $\delta^{18}O$ and winter Greenland $\delta^{18}O$ PC1 are obscured in the annual

234 correlation with Greenland $\delta^{18}O$ PC1 due to the dominance of summer accumulation (Masson-Delmotte et al.,

235 2015).

**6 Comparison with regional climate**

**6.1 Association with the temperature and atmospheric circulation**

238 Fig. 5a and 5b show the spatial correlation maps between NEEM seasonal $\delta^{18}O$ and surface air

239 temperature (SAT) retrieved from the 20CR data set. All data are detrended before correlation. NEEM summer

240 $\delta^{18}O$ is significantly positively correlated with May-Oct averaged SAT over all of Greenland, Baffin Bay and the

241 open water to the east of Greenland. This significant correlation also occurs as far south as 35°N in the North

242 Atlantic where a previous study suggests the possible moisture source for precipitation at NEEM (Steen-Larsen

243 et al., 2011). For winter $\delta^{18}O$ and Nov-Apr averaged SAT, no correlation is displayed over Greenland or nearby

244 consistent with the results from observations. Winter $\delta^{18}O$ correlates significantly with the SAT near 35°N in the

245 North Atlantic and the Canadian Archipelago. But the correlation coefficients are only up to 0.25. Due to less

246 reliable data in the early stage of 20CR data, we also examine the correlations within two sub-intervals 1855-

247 1930 and 1931-2004 (Fig. S3). The strong extended correlations between NEEM summer $\delta^{18}O$ and May-Oct

248 averaged SAT are consistent within two sub-intervals. For winter correlations, both show no correlations over

249 Greenland or nearby. The correlations with the SAT data from the reanalysis data support the conclusion that

250 summer $\delta^{18}O$ from NEEM has a better correlation with temperature than winter $\delta^{18}O$.

251 The NEEM seasonal $\delta^{18}O$ is also compared with the sea level pressure (SLP) from 20CR data for the

252 same time intervals as temperature (Fig. 5c, 5d and Fig. S4, supplementary information). There is no obvious

253 NAO-like pattern (the seesaw structure over the North Atlantic Ocean) for the comparison between summer $\delta^{18}O$

254 and May-Oct averaged SLP for the whole period. A NAO-like pattern emerges for the second sub-period 1931-

255 2004 (Fig S4b), but the northern node is limited suggesting a rather weak summer NAO footprint on $\delta^{18}O$ at

256 NEEM. There is a seesaw structure when correlating winter $\delta^{18}O$ with Nov-Apr averaged SLP over the last 150

257 yrs (Fig. 5d) and within the subperiod 1855-1930 (Fig. S4c). However, the correlations with SLP are also rather

258 weak for these periods. The absolute values of correlation coefficients are less than 0.33 both for 1855-1930 and

259 for whole period. Furthermore, it should be noted that there is an absence of the NAO-like pattern for the second

260 75-year period (Fig. S4d) when observations are generally more reliable due to the increased number of

261 assembled observations around Greenland. Hence, care should be taken when interpreting inconsistent

262 correlations in the sub-intervals. Another interesting feature in Fig. S4c and S4d is the consistent negative

263 correlation between NEEM winter $\delta^{18}O$ and Nov-Apr averaged SLP over North America and Canadian

264 Archipelago within the two subperiods. This suggests that NEEM winter $\delta^{18}O$ is more likely influenced by the

265 pressure over North America and Canadian Archipelago.



As the circulation indices are the simplified indicators of circulation patterns, we here further investigate
the possible connections to AMO and NAO patterns with the seasonal NEEM data (Table 1). Both indices (Fig.
4e and f) and NEEM seasonal data are detrended before correlation. The summer $\delta^{18}O$ signal correlates weakly
with May-Oct averaged AMO (r=0.22) over 1855-2004. The correlations are also consistent within the two-
subintervals. There is no correlation between winter $\delta^{18}O$ and Nov-Apr averaged AMO. Summer $\delta^{18}O$ correlates
weakly with May-Oct averaged NAO over the whole period and for 1931-2004 but no correlation is seen in the
1855-1930 period. It should be noted that for the whole period the summer correlation with NAO is significant,
but no NAO-like pattern is seen in correlation with SLP for 1855-2004 (Fig. 5c). This may be attributed to the
rather weak correlation with NAO which only is -0.16. NEEM winter $\delta^{18}O$ has no correlation with the Nov-Apr
averaged NAO in 1931-2004. Although the correlation map with SLP shows NAO-like pattern for the 1855-
1930 and the 1855-2004 period, the correlation coefficients with Nov-Apr averaged NAO indices are also rather
weak (r=0.217 for 1855-1930 and r=0.191 for 1855-2004). Furthermore, it should be noted that even if there are
correlations between seasonal NEEM $\delta^{18}O$ and AMO and NAO, those circulation patterns can only explain less
than 7% of the variance of NEEM $\delta^{18}O$. We conclude that both patterns exert weak influence on NEEM $\delta^{18}O$
even do correlations between seasonal circulation indices and seasonal NEEM $\delta^{18}O$. The weak correlations with
NEEM $\delta^{18}O$ are likely due to a larger distance from the Atlantic Ocean and a much lower snow accumulation at
NEEM than other ice cores in central and southern Greenland (Chylek et al., 2012;Steen-Larsen et al., 2011).
The weak correlations can also explain why NEEM annual $\delta^{18}O$ is highly correlated with annual Greenland $\delta^{18}O$
PC1, but surprisingly weakly correlated with annual and winter NAO (Masson-Delmotte et al., 2015) which
leave a strong footprint in most ice cores in central and southern Greenland (Vinther et al., 2003;Vinther et al.,
2010). The seasonal precipitation bias at NEEM which is dominated by summer precipitation, skews the NEEM
annual average $\delta^{18}O$ towards summer. Therefore, the NEEM annual $\delta^{18}O$ presents a summer-biased signal which
has strong correlation with Greenland $\delta^{18}O$ PC1. Furthermore, irrespective of the weaker winter signal in the
annual $\delta^{18}O$, we also find that the isolated NEEM winter $\delta^{18}O$ correlates poorly with winter NAO. This weak
correlation between winter NEEM $\delta^{18}O$ and winter NAO is in contrast with the finding of a strong winter NAO
footprint in the winter $\delta^{18}O$ records in central/southern Greenland. This is important to know when considering
NEEM $\delta^{18}O$ for use in circulation reconstructions using emerging re-analysis techniques (e.g. Hakim et al.,
2016), where a strong seasonality can both be a caveat, but also be exploited for climate reconstructions.
**6.2 Comparison with sea ice concentration**
In this section, NEEM seasonal $\delta^{18}O$ is compared with the SIC record in Baffin Bay for 1901-2004 (Fig.
6). The period is further divided into prior satellite observation period (1901-1978) and satellite observation
period (1979-2004) for comparison. The year 1979 is the onset year of the satellite observations which is
regarded as the more reliable data source. Prior to the satellite period, the data are mainly calculated by the
compilation of historical data (Walsh and Chapman, 2001). SIC data are linearly detrended before correlations
(Fig. 4g). The NEEM winter $\delta^{18}O$ correlates significantly with Nov-Apr averaged SIC extent over Baffin Bay in
1979-2004 with correlation coefficients of up to -0.62 (Fig. 6d). The correlation coefficient between NEEM
winter $\delta^{18}O$ and averaged SIC over the whole Baffin Bay is -0.53. Prior to the satellite period the correlation
between NEEM winter $\delta^{18}O$ and averaged SIC over Baffin Bay is also significant (r=-0.27). Summer $\delta^{18}O$
correlates well with May-Oct averaged SIC in 1979-2004 with correlation coefficients of up to -0.59 along the
Greenland western coastal area (Fig. 6b). The correlation between NEEM summer $\delta^{18}O$ and averaged SIC over

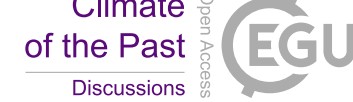

Baffin Bay is also significant (r=-0.46). However, in contrast to the good correlation in the late 20th century,
there are limited significant correlations over the southern part of Baffin Bay for summer in the 1901-1978
period. There is no correlation between NEEM summer $\delta^{18}O$ and averaged SIC over Baffin Bay (r=-0.04) for
this period. One possible explanation for the weaker correlations both for winter and summer in 1901-1978 may
be due to less reliable historical data sources. Furthermore, the reconstructed summer SIC can be underestimated
sometimes due the lower concentration along the coastlines (Titchner and Rayner, 2014). The correlations with
SIC in 1901-1978 are expected to be re-examined in the future possibly leading to an improved sea ice dataset
(Titchner and Rayner, 2014). But still both winter and summer $\delta^{18}O$ are strongly negatively correlated with
Baffin Bay ice extent for 1979-2004 sharing more than 22% variance, which is in agreement with the
relationship between the annual Baffin Bay sea ice anomaly and NEEM annual $\delta^{18}O$ data as illustrated in Steen-
Larsen et al. (2011).
A possible explanation for the sea ice effect on $\delta^{18}O$ is that a reduced sea ice cover may amplify regional
temperature changes and favor enhanced storminess and enhanced precipitation (Noël et al., 2014;Sime et al.,
2013) thus bringing more local moisture. By contrast to the long-distance transport of moisture from the North
Atlantic, the local source leads to less depleted $\delta^{18}O$ in the clouds and thereby, increases NEEM $\delta^{18}O$. However,
this mechanism cannot explain the good correlation with winter $\delta^{18}O$ as NEEM winter $\delta^{18}O$ is poorly correlated
with SAT over the Baffin Bay (Fig. 5b). One hypothesis of this significant winter correlation with SIC may be
attributed to the wind over Baffin Bay. However, we find no correlations between NEEM winter $\delta^{18}O$ and Nov-
Apr averaged wind speed/direction at 850mb and 200mb altitude (jet stream) over 1901-1978 and 1979-2004
(not shown), which may exclude this hypothesis. Another possible hypothesis for the winter correlation could be
the climatic connection between sea ice extent and temperatures in clouds affecting the isotopic composition of
the moisture (Steen-Larsen et al., 2011). Future work can focus on investigating the possible driving factors for
this strong winter correlation which is also consistently significant for the early 20th century. The strong
correlations with SIC indicate the possible strong influence of sea ice changes on the variability of stable isotope
ratios in northern Greenland. It was found that high $\delta^{18}O$ values during the last inter-glacial period (the Eemian
period) could not be achieved in interglacial simulations driven by orbital forcing alone (Sime et al., 2013). Sime
et al. (2013) suggest that sea ice reduction may be the most likely cause of high interglacial $\delta^{18}O$ in Greenland
ice cores. This explanation is supported by our study showing that changes in SSTs and sea ice cover are indeed
key to understanding the past changes in Greenland water isotopes.

**7 Conclusion**
The climate signals archived in stable isotopes in ice cores are complex and can be difficult to disentangle
with annual isotope data only, especially for the NEEM ice core with uneven seasonal accumulation. Combining
four NEEM shallow ice cores, we extracted the seasonal $\delta^{18}O$ signals at NEEM over the 1855-2004 period,
identifying 30% and 70% of the annual accumulation being representative for winter and summer precipitation,
respectively. The quantifications of the signal to noise ratios indicate that a robust seasonal signal can be
extracted from 4 parallel ice cores at NEEM.





NEEM summer $\delta^{18}$O is closely associated with Greenland temperatures. Correlation analysis with 20CR
temperature data indicates strong correlations over the whole of Greenland, the Baffin Bay, and areas as far
south as 35° N. NEEM winter $\delta^{18}$O shows no correlation with Greenland temperatures. The NEEM summer $\delta^{18}$O
record, rather than NEEM winter $\delta^{18}$O or NEEM annual average $\delta^{18}$O, has been shown to be the better
temperature proxy in Northwestern Greenland. The NEEM summer $\delta^{18}$O variability is coherent with the
Greenland summer $\delta^{18}$O PC1 (sharing up to 30% variance) while the winter signal is not, which indicate a
seasonal shift in the impact of circulation and large differences in the regional climate signal in Greenland. The
good summer correlations with temperature and Greenland $\delta^{18}$O PC1 agree well with annual correlations which
are however, dominated by the large fraction of summer accumulation. While the strong correlations are not
observed in winter signal. We conclude that the annual $\delta^{18}$O signal is dominated by summer signal at NEEM
where summer precipitation is dominant. At such seasonally precipitation biased sites it is highly desirable to
identify the seasonal $\delta^{18}$O signal even though multiple cores are usually required to minimize the noise.
Despite the dominant signals of both NAO and AMO in the southern-central ice core isotope data, we
find that, both these circulation patterns exert only a weak influence on seasonal $\delta^{18}$O variations at NEEM. This
has to be kept in mind when combing NEEM $\delta^{18}$O records with other proxy data in circulation reconstructions.
Furthermore, we identify a connection between SIC in Baffin Bay and NEEM summer and winter $\delta^{18}$O in
the satellite SIC data. NEEM winter $\delta^{18}$O shows consistent significant correlations to SIC prior and during the
satellite observation period. This indicates that the NEEM winter $\delta^{18}$O rather than representing temperature itself,
is reflecting sea ice variations and therefore, the distance to the moisture source region. This also opens up for
the possibility of estimating the winter Baffin Bay sea ice extent prior to the onset of satellite observations in
1979 using NEEM winter $\delta^{18}$O.





**Acknowledge**
This work is supported by the scholarship from China Scholarship Council (CSC) under the Grant CSC
No.201606710087. Florian Adolphi was supported by the Swedish Research Council (Grant number VR 4.1-
2016-00218). Raimund Muscheler was also supported by the Swedish Research (Grant Number DNR2013-8421)

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





**Table 1.** The correlations of seasonal NEEM $\delta^{18}O$ records with seasonal averaged different atmospheric circulation indices. The bold text is significant at 95 % confidence level, and the text marked with underline is significant at 90% confidence level (T-test).

| Time | Correlation | | | |
|------|-------------|--------|--------|--------|
| | NAO | | AMO | |
| | winter | summer | winter | summer |
| 1855-1930 | 0.217 | -0.094 | -0.148 | **0.247** |
| 1931-2004 | 0.059 | **-0.252** | 0.148 | **0.255** |
| 1855-2004 | **0.191** | **-0.161** | -0.053 | **0.221** |



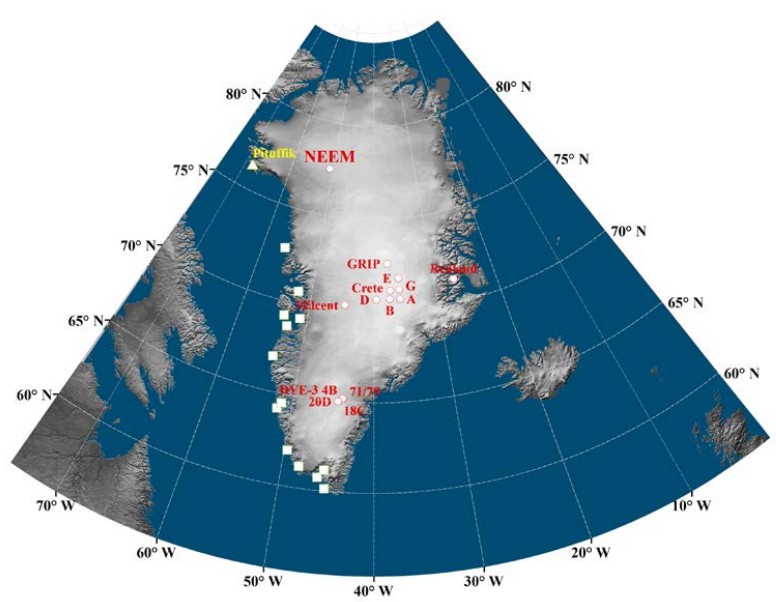

Figure 1. The map of Greenland and ice cores and meteorological stations used for this study. The square indicates the meteorological stations using for SW Greenland temperature series. The Pituffik station is marked as triangle. The ice core sites are shown as circle.



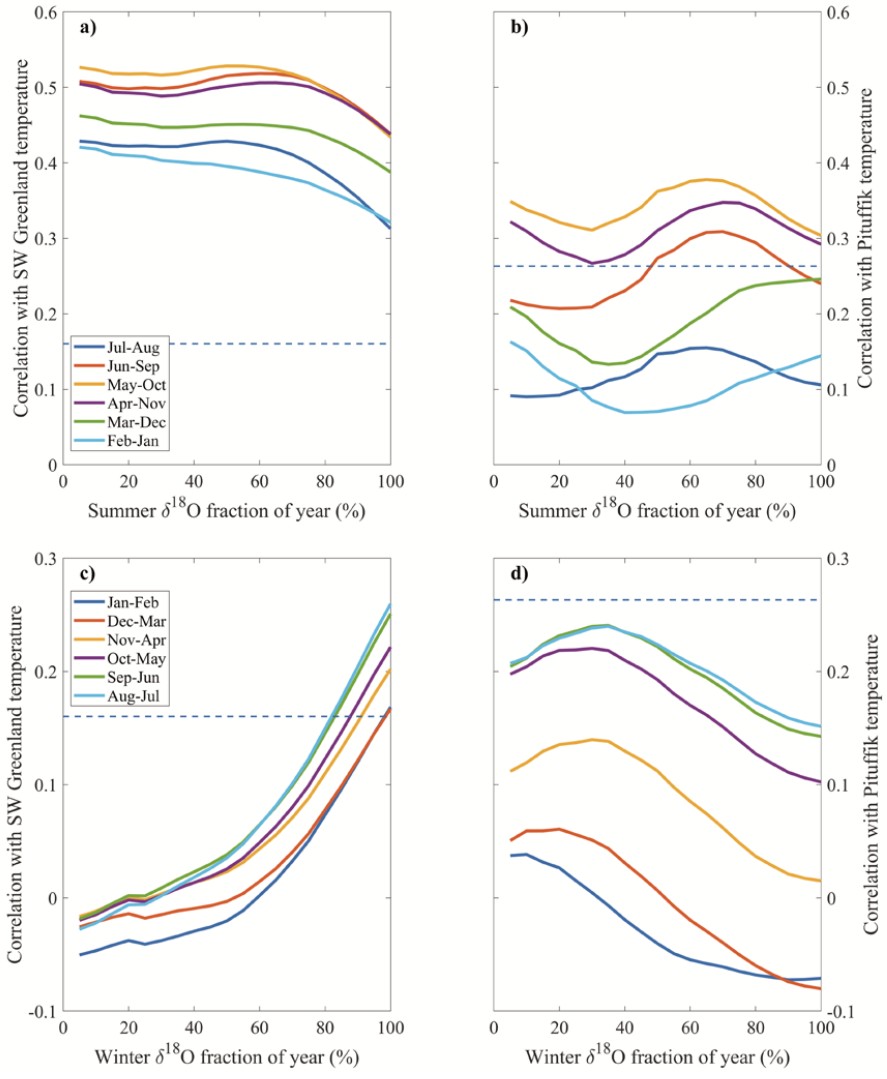

**Figure 2.** Correlation coefficients between stacked data of seasonal δ¹⁸O and SW Greenland (a,c) and Pituffik (b,d) measurement temperature records depending on variously defined choices of seasonal δ¹⁸O data. The analysis covers 1855-2004 for SW Greenland record and the period 1949-2004 for Pituffik record. The 95% confidence level is marked as dashed line (T-test).





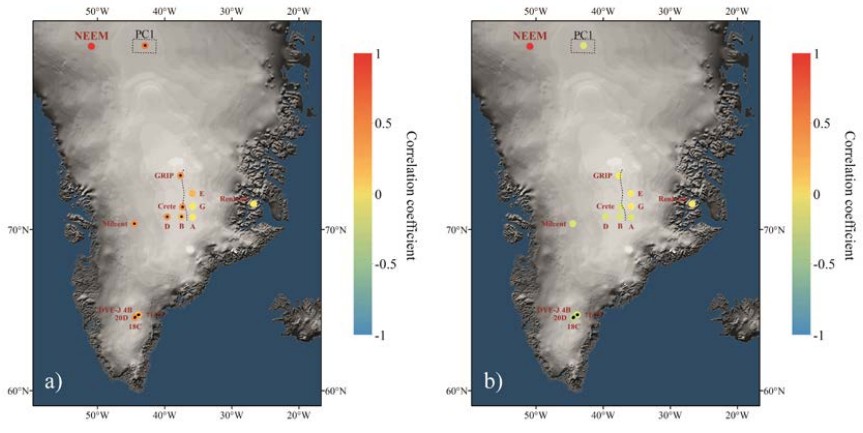

**Figure 3. Correlation coefficients between NEEM seasonal δ$^{18}$O and Greenland seasonal δ$^{18}$O records for the period 1855-1970 (a for summer and b for winter). The PC1 of seasonal central/southern Greenland δ$^{18}$O records is shown within the black dashed rectangle. The ice divide is marked by dotted black line. The significant correlations at 95% confidence level are filled with black dot (T-test).**





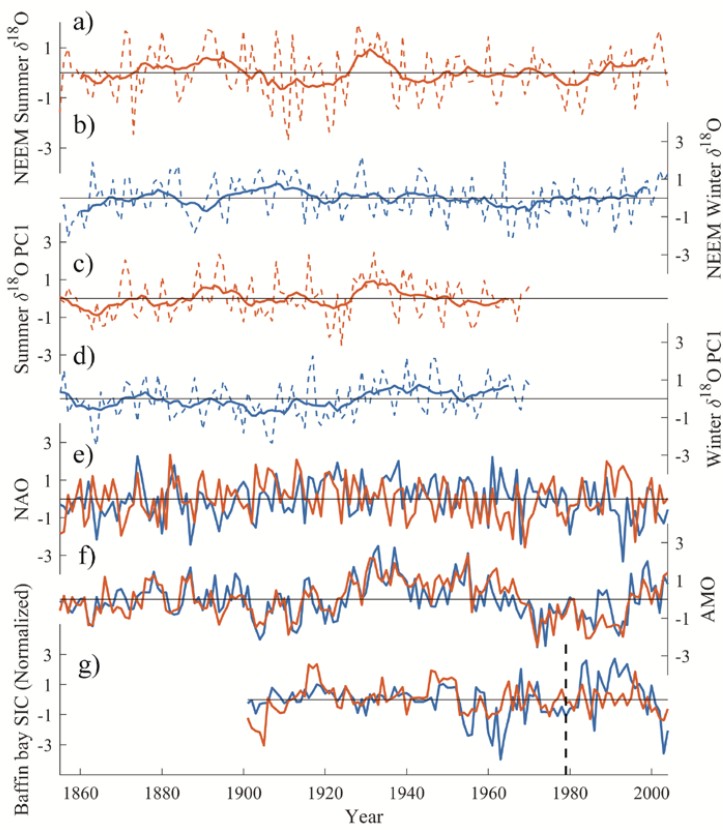

Figure 4. a-b) The NEEM seasonal δ¹⁸O identified in this study. The dashed line and bold line show annual and 11-year averaged data, respectively. c-d) The Greenland seasonal δ¹⁸O PC1 extracted from ice cores in Central/Southern Greenland. The dashed line and bold line show annual and 11-year averaged data. e) The NAO indices calculated from 20CR reanalysis data using principal component analysis. f) The AMO indices calculated from 20CR reanalysis data based on the method by Trenberth and Shea (2006). g) The averaged SIC over Baffin Bay extracted from COBEsic. The dashed line indicates the start year of satellite observation (1979). All red color lines show summer and blue color lines for winter. All data are normalized and detrended.





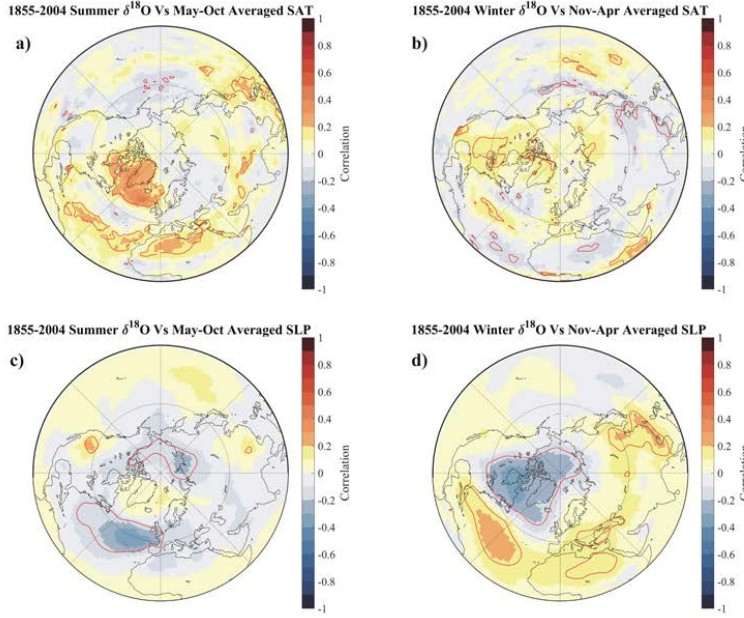

**Figure 5. The spatial correlation map between NEEM seasonal $\delta^{18}$O and SAT (a,b) and SLP (c,d) from 20CR reanalysis data for the period 1855-2004. The winter data are averaged for Nov-Apr and the summer data are averaged for May-Oct. The red solid lines indicate significant correlation at 95% confidence level (T-test)**





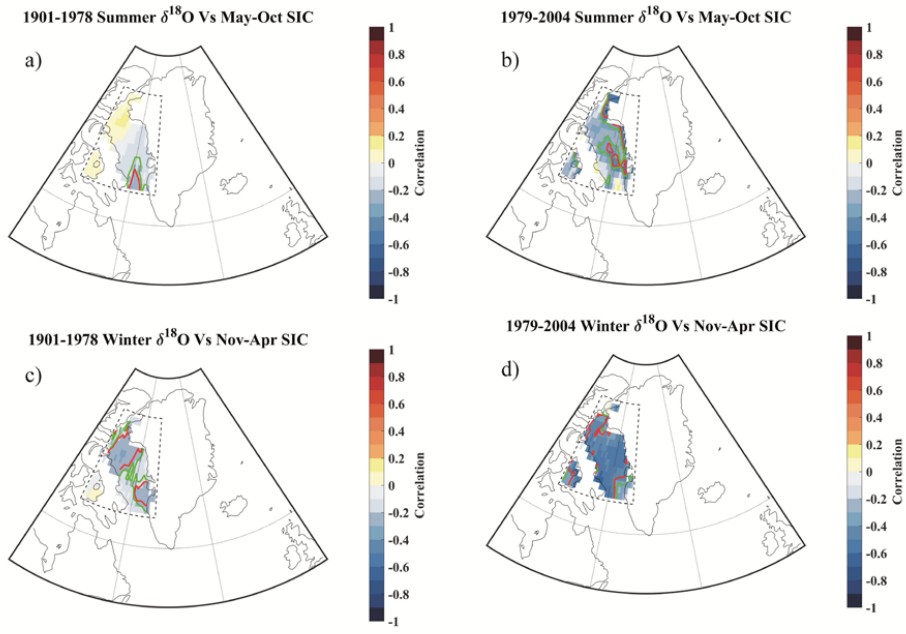

**Figure 6. The spatial correlation map between NEEM seasonal δ¹⁸O and SIC over Baffin Bay for prior satellite observation period (a,c) and satellite observation period (b,d). The winter data are averaged for Nov-Apr and the summer data are averaged for the May-Oct. The red solid lines indicate significant correlation at 95% confidence level and green solid lines at 90% confidence level (T-test).**