# Peer review of "Climate information preserved in seasonal water isotope at NEEM: relationships with temperature, circulation and sea ice"

_Climate of the Past, 2018_

## Referee Comment (RC1) · Anonymous Referee #1 · 30 Apr 2018

Zheng and co-authors are presenting a climate data analysis using d18O records obtained from four shallow firn cores resolved at seasonal resolution drilled at the NEEM Greenland site. They investigate the d18O summer and winter signals, following a previous approach used by Vinther et al (2010) in southern and central Greenland for extracting them. They correlate the seasonal signals to observed meteorological temperatures (summer and winter) recorded in NW and SW Greenland, to twenty century re-analysis data set (20CR), to previously obtained seasonal isotopic records from central and southern Greenland and their PC1, to the NAO and AMO indices. They found that the summer d18O is a better temperature proxy due to a higher snow accumulation rate during summer at NEEM compared to other central and southern Greenland sites.

[Figure]

Moreover, they found a good correlation between winter d18O and sea ice concentration in the Baffin Bay. This last correlation between d18O and sea ice was already suggested by Steen-Larsen and co-authors in a previous paper in 2011 for the same NEEM site but based on interannual variability. However, in this paper, Zheng and co-authors are using for the first-time NEEM seasonal isotopic data rather than mean annual values.

The main outcomes of this paper, the annual d18O dominated by a summer signal at NEEM, the weak influence of NAO and a sea ice climate control during winter compared to other Greenland sites, have important implications for the climate interpretation of the NEEM deep ice core.

The paper is interesting, quite concise, well-structured and the topic is appropriate for Climate of the Past. Nevertheless, the authors should consider some minor comments reported below before resubmitting a revised version.

Page 2, line 62: " . . .. 16 annual resolved . . ..: is this number 13 or 16?

Page 6, lines 207-208: may you expand a little this part, explaining briefly the methodology used here?

Page 6, line 110: " . . .windier conditions and less snow accumulation rate." Are you referring here to wind redistribution phenomena? And/or wind erosion?

Page 6, line 215: comparison with other Greenland ice core records: are there any data from NGRIP? Never mentioned . . ..

Page 9, lines 320-323 and then 325-327: may you explain better here, not clear what you want to say. I do not understand the two different hypotheses.

Page 17, Figure 3: this figure is hardly readable. I would suggest improving it.

Page 18, Figure 4: please add the reference for the data shown in the panel c and d.

---

## Referee Comment (RC2) · Anonymous Referee #2 · 9 May 2018

The manuscript represents a new insight into the factors controlling the formation of stable water isotope content in the north-west Greenland (NEEM site). The authors use 4 shallow cores to study the variability of stable isotopes in snow in 1855-2004. The authors manage to separate the whole ice sequence into summer and winter layers to study differently the corresponding seasonal isotopic signal. They show that summer precipitation constitutes about 70 % of the annual sum. The authors further demonstrate a strong correlation of the summer d18O with the regional summer temperature, but it is not the case for winter d18O and winter temperature. The regional climatic indices (NAO and AMO) are shown to have relatively weak influence on the NEEM isotopes, in contrast to the central and southern Greenland sites. Finally, it is

suggested that winter d18O values in NEEM are primarily governed by sea ice con-centration in Baffin Bay.

The manuscript is nicely written and easy to read. It provides new valuable information and understanding of the processes of the formation of stable water isotopes in the polar regions. Overall, I suggest to publish it with only minor corrections.

Specific comments.

The figures are too small, especially 1 and 3.

Figure 1: you often mention Baffin Bay in the manuscript, it would be nice to show it on the map.

lines 149-150: the average accumulation rate at NEEM (21.6 cm/yr) is only slightly higher than the threshold (20 cm), and since the accumulation rate is highly variable in time, there were periods when it was <20 cm. How does it affect the interpretation of the isotopic record?

lines 151-155 and farther: I understand your way to define the lengths of the seasons and I do not have objections to using this approach. But there are more simple ways to deal with it. You may, for example, divide each annual d18O cycle into winter and summer halves using a chosen d18O value, and then define for each year the amount of ice accumulated during summer and winter, as well as mean summer and winter d18O values. Would not it be more straightforward? Could you please comment on this?

line 183: I suggest to write "or a combination of the both, " to make it clearer.

line 209: are these SNR values for a single core or for the stack of 4 cores?

---

## Author Comment (AC1) · 23 May 2018

First, we would like to thank both reviewers for their insightful and helpful comments. Below we will reply each comment point by point, showing the reviewers comments in black and our response in blue. Changes to the original manuscript are highlighted in **bold**.

------------------# Reviewer 1----------------------------

Zheng and co-authors are presenting a climate data analysis using d18O records obtained from four shallow firn cores resolved at seasonal resolution drilled at the NEEM Greenland site. They investigate the d18O summer and winter signals, following a previous approach used by Vinther et al (2010) in southern and central Greenland for extracting them. They correlate the seasonal signals to observed meteorological temperatures (summer and winter) recorded in NW and SW Greenland, to twenty century re-analysis data set (20CR), to previously obtained seasonal isotopic records from central and southern Greenland and their PC1, to the NAO and AMO indices. They found that the summer d18O is a better temperature proxy due to a higher snow accumulation rate during summer at NEEM compared to other central and southern Greenland sites.

Moreover, they found a good correlation between winter d18O and sea ice concentration in the Baffin Bay. This last correlation between d18O and sea ice was already suggested by Steen-Larsen and co-authors in a previous paper in 2011 for the same NEEM site but based on interannual variability. However, in this paper, Zheng and co-authors are using for the first-time NEEM seasonal isotopic data rather than mean annual values.

The main outcomes of this paper, the annual d18O dominated by a summer signal at NEEM, the weak influence of NAO and a sea ice climate control during winter com- pared to other Greenland sites, have important implications for the climate interpretation of the NEEM deep ice core.

The paper is interesting, quite concise, well-structured and the topic is appropriate for Climate of the Past. Nevertheless, the authors should consider some minor comments reported below before resubmitting a revised version.

> We would like to thank the reviewer for the positive evaluation and the good summary of the manuscript.

Specific comments
1. Page 2, line 62: "… 16 annual resolved… is this number 13 or 16?

> The number here is 16. Masson-Delmotte et al. (2015) use 16 annual-resolved $\delta^{18}$O records to do the analysis. While there are only 13 seasonal-resolved $\delta^{18}$O records from Vinther et al (2010). To make this more clear, we changed the Page2 line 48 to
>
> "Vinther et al. (2010) extracted the seasonal $\delta^{18}$O from **13 sites** in central and southern Greenland."

2. Page 6, lines 207-208: may you expand a little this part, explaining briefly the methodology used here?

We changed the text to make it more clear, and now include the definition of SNR used here.

"…. it is important to examine the mean signal to noise variance ratio (SNR) of four seasonal $\delta^{18}O$ series. **The SNR can be calculated as (more details can be found in Vinther et al. (2006))**

$$SNR = \frac{V_a - \frac{1}{N}\overline{V_\iota}}{\overline{V_\iota} - V_a}$$

**Here $\overline{V_\iota}$ is the mean variance of the records going into this analysis, N is the number of records and $V_a$ is the variance of the average record.**"

3. Page 6, line 210: " … windier conditions and less snow accumulation rate." Are you referring here to wind redistribution phenomena? And/or wind erosion?

Yes, we are referring to wind redistribution of snow. To clarify this we changed the line 210 to

"The winter $\delta^{18}O$ is more strongly influenced by noise than the summer signal possibly due to **windier conditions that lead to a more disturbed signal by sastrugi formation and less snow accumulation than during summer.**"

4. Page 6, line 215: comparison with other Greenland ice core records: are there any data from NGRIP? Never mentioned.

There is no seasonal NGRIP $\delta^{18}O$ records due to the low accumulation rate at NGRIP. To make this more clearer, we changed on page 6 line 215 to "**Comparison with other seasonal Greenland ice core records**".

We also added the sentence after Page 4 line 134 to clarify it:
" The NEEM seasonal $\delta^{18}O$ data are also compared with other seasonal records obtained from 13 sites in central and southern Greenland over the period 1778-1970 (Fig.1; Vinther et al., 2010). **There are no other seasonal $\delta^{18}O$ records from northern Greenland.** Most records originate from single ice core while some are stacked records from multiple cores (GRIP, n=6; DYE3-71/79, n=2)."

5. Page 9, lines 320-323 and then 325-327: may you explain better here, not clear what you want to say. I do not understand the two different hypotheses.

We added a sentence after page 9 lines 320-323 to explain this better:
"One hypothesis of this significant winter correlation with SIC may be attributed to the wind over Baffin Bay. **Changes in the wind strength/direction over the Baffin Bay may modulate the moisture transport from Baffin Bay to the NEEM site.**

We also changed page 9 lines 325-327 to:
**"Another possible hypothesis could be that, instead of the direct coupling of precipitation to local moisture sources at NEEM resulting in the high winter correlation, it is merely a climatic connection between sea ice extend and the clouds temperature thereby influencing the isotopic composition of the precipitation at NEEM (Steen-Larsen et al., 2011)."**

6. Page 17, Figure 3: this figure is hardly readable. I would suggest improving it.

The figure is modified with bigger labels.

7. Page 18, Figure 4: please add the reference for the data shown in the panel c and d.

We added the reference (Vinther et al., 2010) for the data shown in the panel c and d.

------------------# Reviewer 2--------------------------

The manuscript represents a new insight into the factors controlling the formation of stable water isotope content in the north-west Greenland (NEEM site). The authors use 4 shallow cores to study the variability of stable isotopes in snow in 1855-2004. The authors manage to separate the whole ice sequence into summer and winter layers to study differently the corresponding seasonal isotopic signal. They show that summer precipitation constitutes about 70 % of the annual sum. The authors further demonstrate a strong correlation of the summer d18O with the regional summer temperature, but it is not the case for winter d18O and winter temperature. The regional climatic indices (NAO and AMO) are shown to have relatively weak influence on the NEEM isotopes, in contrast to the central and southern Greenland sites. Finally, it is suggested that winter d18O values in NEEM are primarily governed by sea ice concentration in Baffin Bay.

The manuscript is nicely written and easy to read. It provides new valuable information and understanding of the processes of the formation of stable water isotopes in the polar regions. Overall, I suggest to publish it with only minor corrections.

We would like to thank the reviewer for the positive evaluation and the nice summary of the paper.

Specific comments.
1. The figures are too small, especially 1 and 3.
   We have modified these figures and increased especially the size of the labels.

2. Figure 1: you often mention Baffin Bay in the manuscript, it would be nice to show it on the map.

   The Baffin Bay label is added in Figure 1.

[Figure]

3. lines 149-150: the average accumulation rate at NEEM (21.6 cm/yr) is only slightly higher than the threshold (20 cm), and since the accumulation rate is highly variable in time, there were periods when it was <20 cm. How does it affect the interpretation of the isotopic record?

   This threshold value is referring to the average accumulation rate (20cm of ice equivalent per year). If the average accumulation rate of an ice core is lower than this value, the seasonal oscillations in the $\delta^{18}O$ data may be obliterated by firn diffusion to a degree that they cannot by

recreated by back-diffusing the data mathematically (Johnsen et al.,2000). We changed the lines 149-150 to make it more clear.

**" (no less than an average accumulation of 20cm ice equivalent per year; Johnsen et al., 2000)"**

4. lines 151-155 and farther: I understand your way to define the lengths of the seasons and I do not have objections to using this approach. But there are more simple ways to deal with it. You may, for example, divide each annual d18O cycle into winter and summer halves using a chosen d18O value, and then define for each year the amount of ice accumulated during summer and winter, as well as mean summer and winter d18O values. Would not it be more straightforward? Could you please comment on this?

We used the method developed by Vinther et al. (2010). It solves two important questions of identifying seasonal $\delta^{18}O$: how much accumulation we should take for summer and winter season and which months are best correlating to the summer $\delta^{18}O$ and winter $\delta^{18}O$. We assumed that the $\delta^{18}O$ data correlates well with temperature and that the $\delta^{18}O$ maximum/minimum corresponds to mid-summer/mid-winter. The only markers in the $\delta^{18}O$ data that can be used for season identification, are the summer maxima and winter minima. It is difficult to choose the $\delta^{18}O$ signal (and attribute to a season) as suggested by the reviewer. We assess different fractions of annual accumulation (centered on the summer maxima and winter minima) in order to avoid involving assumptions on the seasonal distribution of snowfall but to rather assess the best possible $\delta^{18}O$ fractions to represent the summer and winter signal. Since we also do not know a priori which months could best represent the chosen $\delta^{18}O$ data, we correlate the chosen seasonal $\delta^{18}O$ records with different chosen months, to finally objectively decide which seasonal $\delta^{18}O$ records correspond best to which seasonal climate (e.g. which months best represent the winter signal).
We think our adopted method solves these two questions well and provides rather good results, provided that there is very limited information on how to best use the $\delta^{18}O$ data to extract the seasonal climate information

5. line 183: I suggest to write "or a combination of the both, " to make it clearer.

The sentence has been changed accordingly.

6. line 209: are these SNR values for a single core or for the stack of 4 cores?

The SNR values are for the 4 cores. We changed the line 209 to make it clearer. See comments to reviewer 1